# Ionizing Radiation-Induced Epigenetic Modifications and Their Relevance to Radiation Protection

**DOI:** 10.3390/ijms21175993

**Published:** 2020-08-20

**Authors:** Mauro Belli, Maria Antonella Tabocchini

**Affiliations:** 1Independent Researcher, formerly: Istituto Superiore di Sanità, 00161 Rome, Italy; mau.belli1@gmail.com; 2National Center for Innovative Technologies in Public Health, Istituto Superiore di Sanità, Viale Regima Elena 299, 00161 Rome, Italy

**Keywords:** ionizing radiation, radiation biology, radiation protection, health effects, epigenetics, low dose radiation, DNA methylation, non-targeted effects

## Abstract

The present system of radiation protection assumes that exposure at low doses and/or low dose-rates leads to health risks linearly related to the dose. They are evaluated by a combination of epidemiological data and radiobiological models. The latter imply that radiation induces deleterious effects via genetic mutation caused by DNA damage with a linear dose-dependence. This picture is challenged by the observation of radiation-induced epigenetic effects (changes in gene expression without altering the DNA sequence) and of non-linear responses, such as non-targeted and adaptive responses, that in turn can be controlled by gene expression networks. Here, we review important aspects of the biological response to ionizing radiation in which epigenetic mechanisms are, or could be, involved, focusing on the possible implications to the low dose issue in radiation protection. We examine in particular radiation-induced cancer, non-cancer diseases and transgenerational (hereditary) effects. We conclude that more realistic models of radiation-induced cancer should include epigenetic contribution, particularly in the initiation and progression phases, while the impact on hereditary risk evaluation is expected to be low. Epigenetic effects are also relevant in the dispute about possible “beneficial” effects at low dose and/or low dose-rate exposures, including those given by the natural background radiation.

## 1. Introduction

There is increasing interest in assessing the robustness of the present system of radiation protection at low doses and/or low dose-rates, typical of those exposures encountered in the workplace, in the environment and in diagnostic medicine (also irradiation of normal tissues in radiotherapy may fall in this type of exposure).

Quantitative evaluation of health risks at these levels of exposure is currently obtained by a combination of epidemiological and radiobiological data and models. Even though no comprehensive and “universal” model of radiation action on living systems, i.e., a model capable of describing all aspects at the different scale involved (molecular, cellular, tissue, organ, organisms), has been developed yet, nevertheless, radiobiology research, after just over a century of existence, has provided a wealth of information on biological response to ionizing radiation. Some important general notions are currently used by international bodies, such as the United Nations Scientific Committee on the Effects of Atomic Radiation (UNSCEAR) and the International Commission on Radiological Protection (ICRP), to extrapolate to low doses and low dose rates the health risk derived from epidemiological data at higher acute doses. These notions are essentially the harmful mutagenic potential of ionizing radiation and its linear dose-dependence at low levels of exposure [1,2]. In particular, the fundamental role of radiation-induced DNA damage in the induction of mutations and chromosome aberrations is currently assumed to provide a framework for the analysis of risks at low radiation doses and low dose-rate exposures [2,3]. Additionally, for the induction of cancer and heritable disease at low doses/low dose-rates, the use of a linear relationship between increments of dose and increased risk is considered a scientifically plausible assumption, even if uncertainties regarding this judgement are recognized [1,2,4,5].

However, non-linear radiobiological responses that can be relevant at low level exposures have been observed for many years, such as the so-called “non-targeted effects” (NTEs), and the (radio) adaptive response (AR). Moreover, it is now well established that ionizing radiation, besides genetic mutations, may also cause epigenetic alterations. In effect, epigenetic events are known to regulate gene activity and expression not only during development and differentiation, but also in response to environmental stimuli, such as ionizing radiation [6,7]. Interestingly, there is evidence that NTEs and AR are inter-related and even more interesting is the possibility that epigenetic mechanisms may have a role in them.

Evidence that such biological phenomena do not fit the classical paradigm of radiobiology, on which the internationally agreed system of radiation protection is currently based [1], has led to much discussion on if and how this paradigm should be modified [8,9,10].

Some excellent reviews have been reported on the historical and methodological aspects of radiobiology paradigm evolution [11] and on the effects of ionizing radiation on DNA methylation [12]. The present review encompasses many important aspects of the biological response to ionizing radiation in which epigenetic mechanisms are shown to be, or could likely be, involved with a focus on the possible implications in health risk assessment at low doses, a key issue in radiation protection.

## 2. The Role of Radiation Biology in Radiation Protection

### 2.1. The Current Paradigm of Radiation Biology

Ionizing radiation is capable of inducing a wide spectrum of DNA alterations, such as: base damage, sugar damage, single strand breaks (SSBs), double strand breaks (DSBs), DNA–DNA and DNA–protein cross-links. Clustered DNA lesions (defined as two or more lesions within one or two helical turns of DNA), such as complex DSBs and non-DSB clustered lesions [13] are considered to be the most biologically relevant form of radiation-induced DNA damage [14,15,16,17,18]. They are expected to be less readily repaired as compared to other radiation-induced damage and to endogenous or metabolism-related cellular damage. Indeed, ionizing radiation is uniquely very efficient at inducing clustered DNA lesions [19]. At low doses, even the passage of a single particle can produce clustered DNA lesions [15,17,20].

The frequency and degree of clustering of DNA damage depend on radiation quality [21]. There is evidence that clustered DNA damage, such as multiple DSB as well as non-DSB lesions close together [22] is the most challenging to repair and that the proportion of clustered damage increases with Linear Energy Transfer (LET), reaching ~70% or more for high-LET radiation (see the review in [23]).

High-LET charged particles typically induce complex chromosome aberrations [24,25] (defined as those aberrations involving three or more breaks in two or more chromosomes [26], although they can also be observed less frequently after exposure to γ-rays. In particular, high-LET heavy ions induce a high fraction of complex-type exchanges, and possibly unique chromosome rearrangements [27,28].

Un-repaired or mis-repaired DNA lesions cause changes in the DNA sequence, i.e., (genetic) mutations, that in turn are considered as the main event leading to deleterious biological effects, resulting, even at low doses, in an increase in both the probability of developing cancers and the rates of hereditary diseases naturally occurring in the population [2].

The association of genetic mutation to detrimental effects dates back to 1926 with Muller’s discovery of mutagenic effects of X-rays or γ-rays on the fruit fly *Drosophila melanogaster*, although they were observed after high doses [29]. Muller, who for this discovery was awarded the 1946 Nobel Prize in medicine and physiology, became convinced that the vast majority of mutations were deleterious and consequently that exposure to radiation should be strictly controlled.

Indeed, it is now generally assumed that a vast majority of mutations are neutral or detrimental, as in many cases gene mutation is a process which burdens the population with a load of harmful genes. On the other hand, mutations may occur that, despite their rarity, increase the fitness of the biological system. However, considering the low likelihood of these favorable mutation events, radiation-induced mutations in humans, even at low doses, are generally assumed to be detrimental for radiation protection purposes [30].

A schematic and rationalized picture of the radiobiological knowledge for radiation protection purposes can be summarized by the following statements, forming what is sometimes referred to as the “conventional paradigm of radiobiology” [8], still considered as an useful reference framework:(i)The DNA damage in directly exposed cells is the main event for biological effects;(ii)the DNA damage occurs during, or very shortly after, irradiation of the nuclei in targeted cells;(iii)the potential for biological consequences can be expressed within one or two cell generations;(iv)at low doses, the biological effect is in direct proportion to the energy deposited in nuclear DNA.

The present internationally agreed system for radiation protection has used this paradigm, although with many simplifications and assumptions [1]. It forms the rational basis for assuming a linear relationship between risk and dose in radiation protection, known as the “Linear No-Threshold” (LNT) assumption.

### 2.2. Challenges to the Current Paradigm

Awareness is presently growing that a number of observations challenges the conventional paradigm, based on the target theory of radiation-induced effects. The most relevant are the occurrence of: (i) radiation induced epigenetic effects, i.e., changes in gene expression, for example through alteration of DNA and chromatin organization without altering DNA sequence; (ii) non-linear responses, such as non-targeted effects, i.e., effects observed in cells not directly traversed by radiation (bystander effects, BE) or occurring in the genome of the progeny of irradiated or bystander cells (genomic instability, GI), and (radio)adaptive responses (AR); all these NTEs can be described as the expression of inter- or intra-cellular signaling and are deemed to be particularly relevant to cell response to low doses.

## 3. Ionizing Radiation Induces Epigenetic Changes

### 3.1. The Main Epigenetic Modifications

By the second half of the last century, it was recognized that DNA by itself does not determine all characteristics of an organism, including the human one. The role emerged of those characteristics that crucially determine which genes are expressed in each cell type (“epigenetic” traits). The term “epigenetics” was coined in 1942 but its contemporary usage is quite recent, and for some years it has been used with variable meanings [31]. The modern definition of epigenetics is “the study of mitotically and/or meiotically heritable changes in gene function that cannot be explained by changes in DNA sequence” and the epigenetic trait (epigenome) of an organism is intended as the “stably heritable phenotype resulting from changes in a chromosome without alterations in the DNA sequence” [32].

Epigenetic events are known to regulate gene activity and expression during development and differentiation. In particular, epigenetic mechanisms regulate the gene expression in our body’s cells to create all the different cell types, although they have the same genome. However, they also affect gene expression in response to environmental stimuli, including ionizing radiation (see the reviews in [6,7,33]). Epigenetics is thus considered to be a bridge between genotype and phenotype. Genetic mechanisms, such as mutations, are heritable, but not very susceptible to, or driven by, environmental influence (even if mutations can be induced by the environmental radiation, they are relatively rare events). At the other extreme, there are the metabolic pathways, responsive to environmental changes through interactions of chemical agents or other stressors with proteins involved in gene expression, that are not heritable. Epigenetic modifications, instead, are susceptible to environmental change and heritable at the same time. An interesting aspect is that they can persist after the stressor is removed, but they can also be reversible [34]. The main epigenetic changes currently considered are DNA methylation, histone modification, and modulation of non-coding RNAs (ncRNAs) (Figure 1) [35].

DNA methylation, i.e., the addition of methyl groups to the DNA. In mammals, DNA methylation is mostly at CpG sites to give 5-methylcytosine (5-mC). These sites are concentrated in specific regions called CpG islands, i.e., DNA sequences with high level of CpG sites (typically 300–3000 bp with C + G content > 50%), sometimes located consecutively. In humans, CpG islands occupy about 70% of human gene promoter regions [36]. In transcriptionally active regions of the genome, GpC islands are normally hypomethylated, allowing that gene to be expressed. Therefore, the methylation of CpG sites is a critical factor affecting gene transcription because of its ability to directly silence gene expression. DNA methylation was one of the first identified and the most widely studied epigenetic alteration [37]. It is now a consolidated notion that hypermethylation of genomic DNA is linked to transcriptional silencing and hypomethylation to chromosomal instability [38,39]. DNA methylation is considered a heritable epigenetic mark since methylation modifications that regulate gene expression are usually heritable in mitotically dividing cells. In contrast, it shows dynamic changes during development and cell differentiation, even if some methylation patterns may be retained as a form of epigenetic memory [40]. In mammals, DNA methylation patterns are maintained or established by a family of enzymes, the DNA methyltransferases (DNMTs), notably DNMT1 (maintenance methylation) and DNMT3 (de novo methylation), while other proteins can achieve active de-methylation ([38] and refs therein).

Histone modifications, including, inter alia, acetylation, methylation, phosphorylation and ubiquitination. For years, histones were regarded as merely structural proteins, but now they are recognized to control the organization of chromatin and hence transcriptional responses [41]. Post-translational modifications on histones can change gene transcription by changing DNA accessibility, but also by recruiting other proteins. Histone acetylation, the first epigenetic modification shown to be connected with biological activity [42], neutralizes histone positive charges and reduces its interaction with the negatively charged DNA, thereby inducing chromatin structure relaxation and a marked increase in gene expression. On the contrary, histone methylation does not alter the charge of the modified residue and can either repress or activate transcription depending on the methylation site [43].

Modulation of non-coding RNAs (ncRNAs). Among these RNAs, much attention has been paid to microRNAs (miRNAs), which are small RNA molecules (usually 21–23 nucleotides) discovered in 1993 [44]. Countless microRNAs have been discovered and described in the past years [45,46]. They play an important role in animals and plants in regulating gene expression by transiently inhibiting the translation of a messenger RNA molecule or by inducing its degradation [47,48]. In addition, long non-coding (lnc) RNA molecules may have an epigenetic role [49], since they bind to the transcripts in the nucleus as they emerge from the DNA. miRNAs are involved in multiple biological processes, including cell proliferation, differentiation, and programmed cell death. Since the dysregulation of these processes is a hallmark of cancer, miRNAs can be viewed as important contributors to the pathogenesis of cancer, including initiation and progression [50,51]. They are estimated to regulate the expression of up to 60% of the human protein coding genes [52,53] by means of mRNA degradation or translational repression, acting through a multitude of interconnecting regulatory pathways [51,54,55,56].

### 3.2. Radiation-Induced Changes in DNA Methylation

Early findings obtained at the end of the 1980s indicate that exposure to ^60^Co γ-radiation causes a dose-dependent decrease in DNA methylation, in terms of levels of 5-mC, in several cultured cell lines [57]. Since then, considerably amount of research carried out both in vitro and in vivo showed that X- or γ-rays can change the DNA methylation pattern (see the reviews in [12] and [58]).

Studies on cultured human cells showed that low-LET radiation induces DNA hypomethylation that displayed different profiles in radioresistant and radiosensitive cultured human cells [59,60]. Animal studies, in particular on rodent models, indicated that low-LET radiation induces global DNA hypomethylation that is not ubiquitous among different tissues and cells [61], that occurs in a dose-dependent, sex-, and tissue-specific manner [62,63], and that can be persistent [64,65].

Overall, these data indicate that low-LET radiation exposure results in global DNA hypomethylation. However, it is important to identify whether or not hypomethylation is uniformly distributed throughout the genome, and whether there is also specific locus hypermethylation, which is known to be associated with inactive chromatin state and in most cases with repressed gene expression activity [66,67,68]. It should be considered that the majority of the eukaryotic genome is composed of repetitive elements (REs), while only less than 2% is occupied by protein-coding genes [69]. Non-coding REs, in particular the so-called transposable elements (TE), provide a rich source of gene regulation. Their hypomethylation, especially in the regions called “Long Interspersed Nucleotide Element 1” (LINE-1), has been observed in virtually all human cancers and is frequently associated with a poor prognosis [70]. Loss of DNA methylation in the TEs enhances transcriptional activity so that reactivation of TEs potentially leads to GI [58,71,72,73,74], considered as a major hallmark of many cancer ([75] and refs therein). Many lines of evidence clearly demonstrate that alterations in methylation and expression of TEs are caused by exposure to environmental stressors, many of which are carcinogens or suspected carcinogens so that it has been proposed that TEs can serve as biomarkers of exposure to environmental stressors [72]. However, hypermethylation of TEs has also been detected in some in vitro experiments, suggesting that alterations in the methylation status of TEs is tissue-, dose-, and radiation quality-dependent (see [72] for a review).

Specific-gene hypermethylation often involves normally unmethylated CpG islands, and can be associated with transcriptional silencing of the corresponding gene. If it is a suppressor gene, its loss of function may be a key event contributing to the oncogenic process [76,77,78,79]. In effect, some studies demonstrated significant DNA hypermethylation of tumor-suppressor genes in workers exposed to ionizing radiation [80,81]. Also, gene-specific DNA methylation changes was found in human breast cancer cells irradiated with X-rays [82]. Interestingly, this differential methylation changes correlate with already known biological responses to radiation, such as those on cell cycle, DNA repair, and apoptosis.

### 3.3. Radiation-Induced Histone Modifications

Cell exposure to ionizing radiation results in a wide variety of histone modifications. A well-known radiation-induced histone modification is phosphorylation of histone H2AX, which is crucially important for the repair of DNA double strand breaks and for the maintenance of genome stability. Phosphorylation of this histone at serine 139 (γ-H2AX) is an early cellular response to ionizing radiation and is used as a measure of DSBs [83,84].

In an in vivo murine model, low-dose X-ray irradiation resulted in decreased tri-methylation of histone H4 in the thymus accompanied by an overall reduction in chromatin compactness, a significant increase in global DNA hypomethylation as well as an accumulation of DNA damage, and was associated to a reduced expression of DNMTs [85]. Similar histone modifications were found in human breast cancers [86]. These findings demonstrate that radiation-induced changes in DNA methylation and histone modifications result in overall GI (see [43] for a review).

Furthermore, it has been shown that chromatin modification by histone acetylation is also crucial for DNA repair [87], and that chromatin acetylation is involved in several important steps such as chromatin remodelling and tagging of DSBs, activation of repair regulators, cell cycle regulation, and apoptosis [88].

### 3.4. Radiation-Induced Modulation of Non-Coding RNA Expression

Another type of epigenetic radiation-induced modification involves ncRNAs, in particular miRNAs that have specific roles in the regulation of gene expression. Since their discovery in 1993 [44], miRNAs have emerged as important modulators in many cellular pathways, including cell proliferation, differentiation, and programmed cell death, and the roles of specific miRNAs have begun to be elucidated.

A number of studies have examined the general and specific effects of miRNA perturbation in different cell types exposed to low-LET ionizing radiation (see [89] for a review). miRNAs have been shown to be involved in the response of irradiated cultured human cells [90]. In particular, it was shown that ionizing radiation affects miRNA levels in human endothelial cells [91]. Overall, these studies revealed that the expression levels of several miRNAs change significantly upon irradiation and indicated a specific role of various miRNAs on cellular radiosensitivity [92]. miRNAs have also shown to have a fundamental role in several radiation-induced cell signaling events, such as those involving cell cycle arrest and cell death (reviewed in [93]). Many studies demonstrated that miRNA expression levels change in response to radiation, and that certain miRNAs alter radiation sensitivity, suggesting they are good potential targets for enhancing the efficacy of cancer radiation therapy [89,94,95,96,97]. Expression levels of a variety of miRNAs after low-LET ionizing radiation were reviewed and listed in [98].

### 3.5. Radiation Quality May Affect Epigenetic Changes

Most research on the impact of radiation exposure on the epigenome has focused on the effects of low-LET X- or γ-rays. In contrast, few studies have assessed the effects of high-LET radiation on the epigenome. Increased interest in the mechanisms underlying biological effects of high-LET radiation was triggered quite recently by the investigation on the health risk posed by the space radiation during manned space missions and to the introduction of high-LET radiation into clinical practice (hadrontherapy). Comparison between epigenetic effects induced by low- and high-LET radiation was addressed in particular by Morgan’s group [58,90]. It is expected that high-LET radiation has the potential for unique effects on the epigenome, given the unique characteristics of its track structure. Indeed, there are now a number of studies showing that exposure to high-LET radiation can result in lasting changes in the total levels of DNA methylation and in the miRNA expression that may be different from those induced by equivalent doses of low-LET radiation [58,90,99,100,101,102].

Some of these studies focused on the effect of high energy and charge (HZE) particles, such as high energy Fe-ions (usually 600–1000 MeV/u, LET 180–150 keV/μm), as they are representative of the most detrimental component of space radiation associated to health risks encountered by astronauts in deep space [103,104]. Comparison between X-ray and high-LET Fe-ions exposures of cultured cells showed that Fe-ions elicited more chromosomal damage and cell killing than X-rays do [90]. Global DNA methylation was affected in a different way, as hypermethylation was found in cultured cells 16–20 doublings after exposure to protons and high-LET Fe-ions in contrast to hypomethylation for cells exposed to X-rays [58,101]. Global DNA hypermethylation was also confirmed after exposure to Fe-ions in a mouse model [100]. Interestingly, high energy protons of relatively low-LET gave an effect similar to that caused by high-LET Fe-ions, suggesting that epigenetic responses to radiation may be based on radiation quality rather than LET [58].

A possible explanation for the difference between sparsely and densely ionizing radiation comes from the possible difference in oxidative stress [58] or from the observations [102] that stable DNA methylation can result at the sites of DNA break repair [105], likely produced with higher yield by densely ionizing radiation.

However, after exposure to high-LET Fe-ions, TE hypomethylation was detected in the same cultured cells that displayed global hypermethylation [58]. In vivo experiments performed on mouse models irradiated with Fe-ions showed a complex picture: hypo- or hyper-methylation in TEs depended on the organ analyzed and on the observation time (see [12] for a review). It was also clarified that DNA hypermethylation of LINE-1 elements found in the lungs of mice irradiated with Fe-ions depended on their evolutionary age [106].

Presently, little information is available on the effect of high-LET radiation on methylation at specific genes. In vitro experiments showed hyper- or hypo-methylation or no changes at promoters of specific loci that are used as biomarkers for the early detection of carcinogenesis [12,58,90,99]. The observed differences are likely related to differences in cell types, doses/dose rates, time of observation, or assay used. In a mouse model irradiated with Fe-ions, an increase in 5-mC content was reported that, however, was not associated with increased DNA methylation in a panel of tumor-suppressor genes frequently hypermethylated and inactivated in lung cancer [100]. Some information has come from human data on exposed workers (as reviewed in [12]). Significant DNA hypermethylation of the cyclin-dependent kinase CDKN2A, and of the DNA methyltransferase MGMT genes was found in the sputum of uranium miners exposed to radon [80]. This analysis was also proposed to predict lung cancer. Another study found high levels of p16 hypermethylation in lung adenocarcinomas from plutonium-exposed workers at the Russian nuclear plant MAYAK [107]. However, these results should be regarded as qualitative, since it is not easy to quantify the high-LET exposure in these cases.

Dependence on radiation quality was also found for effects on miRNA expression. In cultured cells, Fe-ions irradiation caused a lower incidence of alteration of miRNA expression levels than X-rays do [90], a quite surprising result given the higher effectiveness of Fe-ions for chromosomal damage and cell killing. Irradiation with high energy protons, γ-rays, or Fe-ions in mouse blood resulted in a radiation type- and dose-specific modifications of a panel of 31 miRNAs [108], so that the extent of miRNA expression signatures derived from mouse blood was proposed as a biomarker for exposure to high-energy protons [109].

## 4. Basic Mechanisms of Radiation-Induced Epigenetic Changes

It is well known that ionizing radiation can cause DNA lesions by direct deposition of energy in the DNA as well as by the indirect action of reactive chemical species formed near the DNA [15,110] and that the spectrum of lesions depends on radiation quality [17,18]. Indirect DNA damage from water free radicals is the most frequent mechanism for low-LET radiation, while direct DNA damage is predominant for high-LET radiation [111,112]. These radicals are formed through the radiolysis of water, the hydroxyl radicals being considered the most damaging among them. In aerobic conditions, these free radicals are converted to reactive oxygen species (ROS) that include free radicals as well as non-free radicals. Organic radicals are also formed, giving rise to peroxyl radicals (strong oxidizing species) and hydroperoxydes (see e.g., [113]). Ionizing radiation can also generate reactive nitrogen species (RNS) through the upregulation of several enzymes, including inducible nitric oxide synthase. Nitric oxide reacts with superoxide radical, generating peroxynitrite, a strong oxidant radical [43]. The yield and spatial distribution of ROS and RNS are strongly modulated by radiation quality as a consequence of the specific track structure of each quality [113]. ROS and RNS can attack DNA resulting in several alterations, including DNA breaks, base damage and destruction of sugars. These lesions, if unrepaired or mis-repaired, may lead to genetic mutations in surviving cells. In this context, particularly relevant are the DNA clustered lesions [22], since they appear to be “highly resistant” to faithful repair (see the review in [23]).

The mechanisms by which ROS are generated by ionizing radiation were studied in some detail in fibroblasts, where it was shown that ROS can be directly generated by radiation exposure and indirectly through the damage of mitochondria. This leads to the activation of the signaling pathway, which sustains an increase in ROS levels by increasing oxidase expression, thereby setting up a cycle of high oxidative stress, i.e., excess of ROS/RNS not compensated by the scavenging mechanisms of the cell [114].

Besides the mutagenic action of ROS and RNS, there is also evidence that oxidative stress has a fundamental role in epigenetic modifications [115,116,117]. Oxidative stress can modify the epigenome by multiple mechanisms, the most important of which involve oxidation of DNA bases and/or mitochondria-mediated changes, with the main target being the CpG sites, especially in the CpG islands.

Among the mechanisms leading to global DNA de-methylation/hypomethylation, an important one is the oxidation of 5 mC to 5-hydroxymethylcytosine (5 hmC), which serves as an intermediate in active DNA demethylation [118,119]. In addition, oxidation of guanine to 8-Oxo-2′-deoxyguanosine (8-oxo-dG) can create mismatches via pairing with A, thus leading to G > T transversion. In addition, 8-oxo-dG can also affect dC methylation by interfering with the ability of DNA to function as a substrate for the DNMTs, inhibiting DNA methylation at nearby C bases [120]. A complete understanding of the effect of 8-oxo-dG is still a matter of study, since it may alter gene expression in multiple ways [121].

Mitochondria also appear to have an important role in radiation-induced global DNA hypomethylation. Dysfunction of mitochondria can affect epigenetic regulation [122]. Mitochondria constitute a major intracellular source of reactive species, as they generate almost 90% of the total number of cellular ROS [123]. High intra-mitochondrial ROS level can damage the mitochondrial DNA, causing global DNA hypomethylation, by decreasing the activity of DNMTs and these changes are transmitted to the progeny of the irradiated cells [124]. These observations suggest that mitochondrial dysfunction can cause oxidative DNA damage and contributes to an altered epigenetic landscape to perpetuate radiation-induced instability [125].

In addition to hypomethylation, ROS can also induce site-specific hypermethylation by different mechanisms, such as catalysis of DNA methylation or upregulation of DNMTs levels, thereby leading to gene silencing [126] (Figure 2).

Oxidative stress also contributes to epigenetic changes by altering the action of ncRNAs, in particular miRNA. However, the interactions between ROS metabolism and miRNA levels appear to be complex. There is evidence that miRNAs are critical regulators of the cellular stress response and thus are responsive to ROS, some of them being themselves able to regulate ROS levels ([127], and refs therein). Analysis of ROS-mediated miRNA expression patterns revealed that the gene locations for epigenetic changes correspond to fragile sites known to be targets of oxidative damage [43].

It is important to note that research on radiation-induced epigenetic mechanisms was initially addressed to DNA methylation as a process capable of modulating gene expression by changing chromatin organization, and subsequently integrated with the roles of histone modifications and changes in miRNA expression as they would act independently. However, there is accumulating evidence of interactions between these different types of epigenetic changes.

Histone modifications can regulate the DNA methylation [128,129]. DNA methylation and histone methylation are tied together in a reinforcing loop [130,131,132,133,134]. Cross-talks between DNA methylation and histone modification were shown at specific gene loci and are present in eukaryotic organisms, although they vary widely, in fungi, plants and animals [135]. DNA methylation can also be regulated by miRNAs targeting DNMTs or critical methylation-related proteins, whereas DNA methylation regulates miRNA expression via hypermethylation or hypomethylation of the promoter-associated CpG islands, thereby achieving a sort of mutual regulation [136,137].

In summary, it appears that radiation-induced oxidative stress is an important player in shaping the epigenetic landscape of the entire genome [127], that is a result of a cross-link between DNA methylation, histone modification and ncRNA (in particular miRNA) expression [137,138].

Interestingly, it appears that the production of oxidizing species that are responsible for inducing DNA damage via indirect effects can also have a role in the damage repair processes via epigenetic changes that enable DNA accessibility to repair enzymes. For example, this can be accomplished through histone modification or replacing canonical histones with histone variants, thereby inducing the needed changes in chromatin structure and dynamics [139].

## 5. Epigenetic Changes Have a Role in Radiation-Induced NTE and AR

In the last three decades, a wealth of investigations have been carried out on NTE, namely BE and GI, discovered between the end of 1980 and the beginning of the 1990s [140,141,142,143,144,145,146], and on radiation-induced AR, discovered even earlier [147], that are phenomena that do not fit into the conventional paradigm.

Radiation-induced GI is an encompassing term which is used to describe the acquisition of an increased rate of alterations within the genome, manifested in the unexposed progeny of irradiated cells [145,146] (Figure 3). Radiation-induced BE describes the ability of cells affected by radiation to convey manifestations of damage to other cells not directly targeted [141,148]. Abscopal, or out-of-field, effects, defined in radiotherapy as radiation-induced effects observed outside the irradiated volume, are currently considered as a special type of BE [149]. Abscopal effects were seen in rodents, such as the induction of profound epigenetic dysregulation in spleen tissue after localized cranial radiation exposure [150], and the increased induction of malignancy in the shielded head (specifically in the brain) of radio-sensitive mice after exposure of the remainder of the body to X-rays [151].

These phenomena challenge the concepts on which the conventional paradigm of radiation biology is based and are potentially relevant for radiation risk assessment, especially at low doses [149,152,153]. They have been seen in many in vitro and in vivo experiments, including experiments with blood samples from irradiated humans [145,146,152].

There is evidence that all these phenomena are inter-related and that they may share some common mechanistic pathways (see, e.g., [153,154,155]). For example, the radiation-induced intercellular signaling cascades, including cytokine production, nitric oxide production and persistent free radical production have the potential to mediate both GI and BE. Indeed, GI was observed in the progeny of unirradiated neighbors of irradiated cells [156].

Most NTE have been observed in vitro, but they can also be relevant in vivo, even if the question remains whether the non-targeted effects demonstrated in vitro can be extrapolated to in vivo situations. In vitro experiments have provided some important insights into the nature of these effects, but in spite of extensive research, their mechanisms remain to be completely understood. An intriguing observation is that, even if NTE and AR have been observed in a variety of cell and tissue types, biological end-points and radiation qualities [152,157,158,159], they have not been universally observed [153,160,161,162].

Epigenetic mechanisms, encompassing DNA methylation, histone modification, and RNA-associated gene silencing, have been shown to be plausible mediators of the mentioned effects. These inter-relationships have stimulated much interest, especially for their possible impact in the risk assessment at low radiation doses and have been the subject of a number of studies [73,90,163]. It is worth noting here that in its latest recommendations, ICRP classifies GI and BE as epigenetic responses to radiation [1].

Indeed, there are many lines of evidence that epigenetic mechanisms have a potential role in GI. An early observation was made on micronuclei induction in cultured cells irradiated with different fluences of alpha-particles, indicating that the target for GI is larger than the cell nucleus [164]. It was shown that GI can occur without the need for genetic alterations as an initiating or perpetuating factor [165,166]. Cells irradiated with low to medium doses exhibited a much larger proportion of GI than mutations from targeted effects, suggesting that instability might arise, rather than from a genetic mutation, through epigenetic mechanisms [152].

All the above observations give support to the idea that epigenetic alterations could be a mechanism of GI induction [167]. Indeed, experimental evidence points to a causal relationship between GI in the exposed animals and the radiation-induced global DNA hypomethylation (see the review in [168]). The fundamental role of DNA methylation in the transmission of GI is clearly demonstrated in embryonic stem cells since the disruption of specific DNMT genes completely eliminates the transmission of GI. Interestingly, this inactivation also protects neighboring cells from indirect induction of GI [169].

A recent review highlighted the link between radiation-induced ROS, DNA hypomethylation and GI and/or AR [170]. Moreover, it has been reported that mitochondrial-derived reactive species can not only cause oxidative DNA damage but also directly affect aberrant changes in 5 mC levels, suggesting a link between radiation-induced genomic instability, epigenetic mechanisms and mitochondrial dysfunction [125].

There is some evidence that, besides DNA methylation, miRNA may also have a role in radiation-induced BE and GI. Concerning BE, an experiment conducted in vivo showed that partial exposure of a mouse body induced a significant upregulation of a specific miRNA in distant lead-shielded liver tissue [150]. Various bystander end-points, such as apoptosis, cell cycle deregulation, and DNA hypomethylation, are shown to be mediated by the altered expression of miRNAs, even if they do not appear to be the primary bystander signaling molecules in the formation of bystander-induced DNA strand breaks [163]. Concerning GI, upregulation of miRNAs was found in directly exposed male mice, leading to hypomethylation of the exposed animals as well as of their unexposed offspring, demonstrating the possibility that they may play a role in the transgenerational epigenetic inheritance of GI [171].

Information about the possible role of epigenetic mechanisms in the AR to ionizing radiation is scarce. AR can be regarded as a quite general phenomenon of biological response as it has been observed in cells, tissues and organisms using various indicators of biological damage after exposure to ionizing radiation and to other stressors. Although in the literature there are a plethora of descriptions about the adapting conditions, and DNA repair and antioxidative mechanisms are among the best described pathways involved in it, the mechanisms underlying AR remain poorly understood [172]. Both intracellular and intercellular signaling (the latter being mainly related to BE) can account for the occurrence of AR. Enhanced efficiency of DSB repair through homologous recombination and a significant increase in gene expression of antioxidant enzymes appear to play a predominant role in the adaptive response (see the review in [9]). An interesting AR model has been developed accordingly [173].

However, the processes by which increasing in DNA repair efficiency and in antioxidant levels would be accomplished by exposure to the priming dose are not clear. It has been proposed that, in order to adapt the gene expression program to the stress situation, and to achieve proper functioning of DNA repair processes, epigenetic processes are involved, notably transient protein acetylation [88]. Furthermore, data on endothelial cells suggest that the radiation-induced changes in miRNAs expression modulate the intrinsic radiosensitivity of these cells in subsequent irradiations [91]. A recent review pointed to the radiation-induced oxidative stress as the source of various processes connected to AR [170], which is consistent with the occurrence of epigenetic mechanisms.

## 6. Epigenetics in Radiation Risk Assessment

### 6.1. Radiation-Induced Cancer

There is large consensus on the fact that cancer, in general, is a disease that results from both genetic and epigenetic changes and several studies pointed to the description of cancer as due to a dysregulated epigenome allowing cellular growth advantage at the expense of the host, with mechanisms involving both genetic mutations and epigenetic modification4s [174,175]. This notion applies not only to solid cancers but also to leukemia, in particular to myeloid leukemia [176]. Dramatic changes in DNA methylation are common in cancer and are considered as an early event in many of them [177,178]. DNA methylation changes appear to be even more frequent events than genetic mutations [179,180]. The global loss of DNA methylation at CpG dinucleotides was the first epigenetic abnormality identified in cancer cells [177,181]. Loss of genome-wide methylation, especially in repetitive elements [77], promotes GI, considered as a major hallmark of cancer [182,183]. For its part, gene hypermethylation, often involving normally unmethylated CpG islands, can be associated with their transcriptional silencing and, if they are suppressor genes, their loss of function may be a key event contributing to the oncogenic process [78,79]. For example, the silencing of the BRCA1 gene by promoter hypermethylation occurs in primary breast and ovarian carcinomas, supporting a role for this tumor suppressor gene in sporadic breast and ovarian tumorigenesis [184]. It has been evaluated that more than 300 genes and gene products are epigenetically altered in various human cancers [185] and a meta-analysis of the altered genes in colorectal cancer reinforces their involvement in tumorigenesis [186].

Additionally, in radiation-induced cancers, a role for tumor suppressor gene hypermethylation has been demonstrated. Silencing of suppressor genes was detected in murine models of radiation-induced lymphoma, in lung tumors of rats induced by exposure to Pu-239, and in human lung adenocarcinoma occurring in workers of the Russian MAYAK plutonium plant ([33,81] and references therein). Interestingly, a study focused on lung carcinoma in radiation-exposed MAYAK workers compared to non-worker controls showed that methylation at one gene (coding for a tumor suppressor protein) occurred more often in carcinomas found in exposed workers than in non-worker controls, with a dose-dependent prevalence [107]. Aberrant hypermethylation was observed in an appreciable fraction of patients with renal cell carcinomas living in radiocontaminated areas after the Chernobyl accident [187]. Significant DNA hypermethylation of tumor suppressor genes was detected in workers exposed to radon in uranium mines, even without detectable cancers [80].

The findings described above indicate that radiation exposure, although normally thought to be pathogenic through DNA damage such as deletions and point mutations [188], may also elevate the cancer risk through epigenetic alteration, resulting in GI increase and/or specific silencing of tumor suppressor genes.

While research in cancer epigenetics was initially focused on DNA methylation abnormalities, particularly on CpG island promoter methylation [189], other players have eventually emerged, a not unexpected result given that probably about 40% of human genes do not contain CpG islands in their promoters [190]. Indeed, besides aberrant DNA methylation, which is one of the most well studied epigenetic changes in cancer cells, it was found that histone modifications and chromatin remodeling are also involved in cancer [191,192].

Next-generation sequencing revealed that more than 50% of human cancers harbor mutations in enzymes that are involved in chromatin organization. [193]. Importantly, aberrant activity of histone-modifying factors may promote cancer development by mis-regulating chromatin structure and activity [194], as frequently found in human leukemias [195].

In recent years, there has been tremendous and growing interest in investigating the role of dysregulation of ncRNAs, notably miRNAs, in normal cellular functions as well as in disease processes. Indeed, less than 2% of the entire human genome encodes proteins, while the majority of it (at least 75%) encodes ncRNAs [69]. There is now emerging evidence that these RNAs are involved in the development and progression of leukemia and cancer [196,197,198].

Alterations in miRNA expression may occur following exposure to several stress-inducing anticancer agents including ionizing radiation, etoposide, and hydrogen peroxide (H_2_O_2_). Dysregulation of a family of miRNA was found in Ptch1 ± mice that are highly susceptible to radiation-induced medulloblastoma [199].

These findings are consistent with the general notion that typically, miRNAs involved in radiation tumorigenesis are dysregulated, and this dysregulation is believed to alter the expression of protein-coding mRNA, thereby favoring uncontrolled tumor cell growth, in some cases by decreasing tumor suppressor expression [200]. miRNA-related epigenetic changes have been proposed to be the “missing link” between radiation exposure, radiation-induced genomic instability, and radiation-induced carcinogenesis [90].

It should be noted that most of the investigations focusing on the relationship between radiation-induced cancer and miRNA changes were obtained using rodent models, while relatively fewer studies have been performed on human cancers. An interesting finding of one of these few is the upregulation of a specific miRNA in breast cancer tissue samples derived from Chernobyl radiation-exposed female clean-up workers [201].

While the occurrence is well established of a relationship between radiation-induced cancers and epigenetic changes, the question can be posed whether these changes are the cause of cell transformation, or rather the consequence of it. Indeed, it is now accepted that epigenetic abnormalities along with genetic alterations are involved in the initiation and progression of cancer [189,202]. For example, it was found that in rat mammary cells, the frequency of initiation (the first step in oncogenesis) induced by γ-rays was much higher than specific locus mutations [203,204,205], and that the observed frequency of radiation-induced GI is considerably higher than that observed for gene mutations at a similar dose, suggesting that the latter is highly unlikely to be the initiating mechanism for GI [145,146,188].

It has also been suggested that a crucial role in such steps is played by epigenetically disrupted stem/progenitor cells [205], a hypothesis consistent with the importance that is now given to cancer stem cells in cancer development and perpetuation [206].

Epigenetic considerations also affect individual susceptibility to radiation-induced cancer. Assessment of individual variability in cancer risk is a key area to address for radiation protection. It is recognized that differences in radiation sensitivity between individuals, or groups, may relate to gender, age at exposure, state of health, genetic and epigenetic make-up, lifestyle, and age attained [207].

### 6.2. Transgenerational Effects

The heritable change in gene expression that is induced by a previous stimulus, such as ionizing radiation, is often described as epigenetic memory. Epigenetic memory is a sort of “footprint” that maintains gene expression states through cell generations without changes in DNA sequence and in the absence of the initial stimulus. Epigenetic memory can be considered over different time scales: cellular and transcriptional memory (mitotically heritable) and transgenerational memory (meiotically heritable) (see, e.g., [208]). In this paper, “transgenerational epigenetic effects” are intended as those effects which arise in the offspring of the irradiated organism and that are not due to the inheritance of DNA mutations through the parental germline, according to the current use in radiation protection issues [209]. Epigenetic variation induced by environmental factors contributes to the phenotypic plasticity and adaptive capacity of various species. The molecular basis of cellular memory is a fascinating topic that has been addressed during the last few decades [40].

In many cases, epigenetic changes have been proven to be stable and can lead to transgenerational heritable changes. In plants and in some animals, such as nematodes, transgenerational epigenetic inheritance is well-documented and relatively common [210]. Many examples have been reported for transgenerational epigenetic effects in which environmental exposures, including ionizing radiation, lead to heritable phenotypic changes that pass through male, female and sometimes both germlines (reviewed in [211]). In mammals, epigenetic patterns are largely erased and then remodeled during germ cell development and early embryonic development (epigenetic reprogramming) [212,213].

The first evidence for a radiation-induced transgenerational effect was reported in 1976 by Luning et al. [214], who showed elevated rates of dominant lethal mutations following intraperitonial injection of male mice with a plutonium salt solution. Afterwards, animal models demonstrated that effects of the parental radiation exposure are transmitted through the germline to the progeny of the irradiated parent [145,146,215].

Radiation-induced transgenerational effects may involve radiation-induced genome instability. Indeed, in vitro data have shown that ionizing radiation can induce genomic instability that can manifest in the progeny of the irradiated cells for many divisions [145] and transgenerational induction of chromosomal instability has also been documented in vivo, notably in irradiated rodents [2,146,216].

Immediately relevant questions are whether the effects are common or rare, and whether they are long-lasting or transient. Indeed, in several species, transgenerational effects have been detected in many generations after the parents were exposed to ionizing radiation, (see the review in [168]. Recent results on vertebrates (zebrafish) show that ionizing radiation-related effects in offspring can be linked to DNA methylation changes, many of which could be associated to pathways involved in cancers and apoptosis, that partly can persist over generations. It has also been suggested that monitoring DNA methylation could serve as a biomarker to provide an indication of ancestral exposure to ionizing radiation [217].

A question especially relevant for radiation protection purposes is whether transgenerational radiation effects occurs in humans. While animal studies show such effects, their occurrence is highly controversial in humans. A high risk of leukemia and birth defects has been reported in the children of fathers who had been exposed to radionuclides in the nuclear reprocessing plants [218] and an increase in minisatellite mutations was found in offspring of various groups living close to the Chernobyl site, to nuclear test sites in Kazakhstan and to the Techa-river region [219,220,221]. However, these findings were not supported by studies in the children of atomic bomb survivors in Hiroshima and Nagasaki [222]. A review published in 2013 of these and other available data concluded that “studies of disease in the offspring of irradiated humans have not so far identified any effects on health, possibly in part a result of lack of statistical power”, and that transgenerational effects of radiation, if any, “may be restricted to relatively short times post-exposure, when in humans conception is likely to be rare” [209]. A subsequent review [223] also considered more recent results from a long-term monitoring by Russian Federation of the children of residents exposed to radionuclides after the Chernobyl accident, which showed an increased prevalence of malignant neoplasms, especially childhood cancer, and other disorders. Based on these findings and on the consideration that the negative results of gene mutations in Hiroshima and Nagasaki might be caused by erroneous methodology, these authors concluded that radiation-induced persistent accumulation of genomic instability may cause various disorders in a further generation in humans [223]. Research has been undertaken using plant and animal systems to understand the mechanisms governing the epigenetic transgenerational effects in organisms exposed chronically to low- doses in Chernobyl and in Fukushima areas [224]. The results so far obtained from laboratory and field studies confirm that DNA methylation might be the key to transfer the response to ionizing radiation from one generation to the next, but more in depth studies are needed, involving other epigenetic mechanisms such as histone modifications and microRNAs, linked to responses at higher levels of biological complexity [225].

If transgenerational effects of radiation were to be demonstrated to apply to humans, it may have implications in radiation protection when estimating the hereditary risks (i.e., the risk of induction of genetic diseases expressed in future generations) of ionizing radiation in human populations. According to the current risk assessment system, they are quantified as the harmful genetic effects on the descendants of those exposed, resulting from the induction of germline mutations and their transmission over generations [1]. This implies that mutation induction in directly exposed cells is regarded as the cause of this risk for humans. Since epidemiological studies have not provided clear evidence of heritable effects of radiation exposure in humans, current estimates for radiation hereditary risk are derived from measured germline mutation frequencies in mice [226]. The underlying rationale is that “experimental studies in plants and animals have demonstrated that radiation can induce hereditary effects, and humans are unlikely to be an exception in this regard” [227]. However, if the results of animal and cellular studies on epigenetic transgenerational destabilization of the genome do apply to human populations, then the hereditary risk could be greater than currently predicted. In this case, the question remains about the magnitude and significance of such an effect in the perspective of radiation protection.

### 6.3. Non Cancer Effects

Manifestations of health effects other than cancer and hereditary diseases have been well known after medium/high doses of ionizing radiation. Within months of Roentgen’s discovery of X-rays, severe adverse effects were reported, such as eye and skin injuries. They were historically termed as “deterministic” effects in contrast to the stochastic cancer and hereditary effects, and later referred to as “tissue reactions” [228]. In general, tissue reactions to high/moderate doses are thought to arise mainly as a consequence of cell killing or functional inactivation, but other non-cytotoxic effects, such as disturbances in molecular cell signaling, also play a crucial role in determining tissue response to radiation. For radiation protection purposes, it is currently assumed that they show a “practical” threshold, defined as the dose required to lead to 1% excess incidence [228], at doses that are well above the levels of exposure typically encountered in the public environment, at work or in diagnostic medical uses of ionizing radiation. Recent results from epidemiological and experimental studies indicate possible increased risks for circulatory diseases, cognitive/neurological effects, and cataracts, not only at high doses but also at doses around 500 mGy and, possibly, even lower. In this section, we will give a glimpse of the role of epigenetics in radiation-induced cognitive and cardiovascular effects and cataract.

#### 6.3.1. Possible Epigenetic Role in Radiation-Induced Cognitive Effects

Clinicians have known for decades that patients subjected to cranial radiotherapy for the control of brain malignancies develop severe and progressive cognitive deficits (see, e.g., [229]). Indeed, many studies of childhood cancer survivors (mainly of leukemia) documented cognitive impairment associated with high-dose (40–50 Gy) cranial irradiation [230]. Quite surprisingly, cognitive impairment was observed in a Swedish group treated for hemangioma in infancy with much lower doses, expressed as a ~50% reduction in high school attendance associated with 100 mGy exposure [231]. *In utero* exposed Japanese atomic bomb survivor data also suggest cognitive impairment at high dose, but no cognitive impairment can be demonstrated in the 0–100 mGy dose range [232,233]. However, the obvious differences in the age-at-exposure values (infancy vs. in utero) make it difficult to draw any meaningful comparison between the two studies [234].

Investigations showing cognitive/behavioral deficits caused by charged particles (relevant for protection against space radiation) in rodent models were carried out to understand the possible limitation to human exploration of our solar system [235,236]. Interestingly, in rats a correlation was recently found between behavioral changes and epigenomic remodeling in the hippocampus [237] and between adverse effects on cognition of space relevant irradiation and epigenetic aberrations consisting in increased levels of the DNA methylating enzymes [238].

#### 6.3.2. Possible Epigenetic Role in Radiation-Induced Cardiovascular Effects

Circulatory disease has been recognized as an important late effect of radiation exposure after the evidence arising from radiotherapeutic experience and epidemiological studies following nuclear and other radiation activities [226]. ICRP has classified circulatory disease as a tissue reaction (a generalized definition of the deterministic effects), with a threshold dose of 0.5 Gy [228]. Several studies addressed the candidate biological mechanisms for the circulatory disease effects of radiation At radiotherapeutic doses > 5 Gy, the cell-killing effect on capillaries and endothelial cells plausibly explains effects on the heart and other parts of the circulatory system [239]. At lower doses (0.5–5 Gy), in humans and in in vivo and in vitro experiments, many inflammatory markers are upregulated long after exposure to radiation, while for doses less than about 0.5 Gy, the balance shifts toward anti-inflammatory effects [240,241]. The involvement of epigenetics, namely demethylation of a gene involved in aging endothelial cells, has been reported as one of the several events that contribute to the eventual development of atherosclerotic plaques after a dose of 10 Gy [242]. Changes in DNA methylation of repetitive elements in the heart tissue have also been observed after the irradiation of mice with 0.1 Gy of protons and 0.5 Gy of ^56^Fe-ions, which are charged particles relevant to space radiation. These changes are dynamic and may vary depending on the time after irradiation, going from early global and repetitive elements-associated DNA hypomethylation to late DNA hypermethylation [243].

#### 6.3.3. Possible Epigenetic Role in Radiation-Induced Cataract

Cataract is a progressive opacification of the crystalline lens of the eye which can determine a decrease in central vision, and is very common in the elderly [244]. It is due to a cumulative physiological response to toxic environmental factors leading to an excessive generation of ROS in the lens epithelium cells and in the superficial lens fiber cells, as well as in the aqueous humor [245]. The main effect of ionizing radiation on the eyes is the onset of posterior cortical and subcapsular cataracts [246], while there is little evidence that nuclear cataracts are radiation-induced [234].

In 2012, ICRP indicated a value of approximately 500 mGy as the threshold for cataract induction by low-LET radiation for acute and fractionated/protracted exposure [228]. This is a value lower by a factor of 10 than that deduced in earlier studies.

Even if it remains unknown exactly how ionizing radiation exposure contributes to opacification [247], epigenetic mechanisms, mainly DNA methylation, have been shown to play an important role in the pathophysiology of numerous ocular diseases [248]. For example, a decreased level of α-crystallin expression in age-related nuclear cataract has been shown, linked to the hypermethylation of the CpG islands in a specific gene promoter [249].

### 6.4. Epigenetics and the Low Dose/Dose Rate Issue

One of the main issues in radiation protection is the assessment of health risks of exposures to ionizing radiation at low doses and/or low dose rates, since these are the levels typically encountered in the workplace, in the environment and in diagnostic medicine, i.e., in exposures with a potential impact in our everyday life [250,251,252]. At these levels, the standard epidemiological approaches cannot give reliable information, so that an integration between epidemiological data and radiobiological studies is required to solve this issue [253].

The term “low dose” has several different interpretations in different contexts. In terms of microdosimetry, it is an absorbed dose such that a single cell or nucleus is very unlikely to be traversed by more than one track, so that the number of affected cells is proportional to the absorbed dose. Since the definition of “unlikely” is subjective, a conservative definition [254] is based on a mean number of 0.2 tracks per cell (or per cell nucleus), meaning that less than 2% of the cells will be subject to traversals by more than one radiation track. This would correspond to a dose of only 0.2 mGy of low-LET radiation [255]. In radiation protection, it is assumed that a low dose is ≤ 100 mGy for acute exposure to low-LET radiation [1,2], corresponding to levels above which no firm evidence exists of increased cancer risks in humans from epidemiological data for sparsely ionizing radiation, and that a low dose-rate is ≤ 5 mGy per hour [256]. More recently, in the framework of the European platform on low dose effects, these are assumed as those where there remains substantial uncertainty on the magnitude of health risk, i.e., ≤100 mGy for low LET radiations when considering cancer risks, and ≤ 500 mGy when considering non-cancer diseases, and low dose rates are assumed as those ≤ 6 mGy/h [253].

While there is little information about the health effects from chronic exposure to low dose-rate radiation, radiobiological studies demonstrated that radiation, when delivered at a slow continuous rate or by fractionation, may have strikingly different effects compared to the same dose delivered acutely. Many data have been collected showing that the biological responses to high and low doses of radiation are not only quantitatively, but also qualitatively, different. For example, the cellular response to DSB induction is substantially different for low compared to high doses, in that low doses are insufficient to induce an efficient DSB repair in vitro [257,258]. Importantly, differences in gene expression profiles have been found, and gene expression changes were established as an early indicator of cellular responses to low-dose radiation in a human myeloid tumor cell line [259]. Subsequently, many other data were accumulated for a variety of biological systems [260,261,262], including human tissue models [263] and human tissue irradiated in vivo, where, however, a considerable individual variability in radiation response was observed [264].

Biological effects usually classified as “beneficial” have been shown after low doses of irradiation; not surprisingly, they are often related to epigenetic mechanisms. In vivo mammalian studies have shown that low doses (up to around 100 mGy) reduce the incidence of spontaneous cancers in mice [265]; other “beneficial” effects, in terms of positive phenotypic changes associated to DNA hypermethylation, have been observed in the offspring of mice when they were irradiated with low doses during early gestation [266]. The study suggested that epigenetic alterations may be the memory system that results in “hormesis” after low doses of ionizing radiation, i.e., in a stimulation that induces a beneficial effect. There have been accumulated many data on hormesis and AR after low dose exposure both in vitro and in vivo [251], and there are several lines of evidence that epigenetic mechanisms can be involved in hormesis-like and life-extending responses in model organisms [267]. It has been shown that chronic low-dose radiation exposure is a more potent inducer of epigenetic effects than acute exposure [268]. Specific gene modulations were observed as a result of chronic low-LET irradiation of mice at low doses [269]. A role of DNA hypermethylation was suggested to be involved in adaptive response induced by chronic low-dose γ-irradiation of human B lymphoblast cells [270].

Both laboratory and field studies have demonstrated changes in overall DNA methylation in organisms exposed chronically to ionizing radiation. An interesting conclusion is that, generally, an elevated chronic level of ionizing radiation induced hypermethylation or methylation pattern changes which could be taken as a response to induce DNA stability [225].

A peculiar aspect of low dose and low dose-rate exposure is that related to the ionizing radiation background. Life has evolved on Earth for about 4 billion years in the presence of the natural background of ionizing radiation, even if it was not always the same as today. Without it, life on Earth could not have existed or would not exist in the present form.

Today, the annual dose due to natural background on average approaches 1 mSv, with cosmic contributions slightly less than the terrestrial one [271]. For human organisms, there is also an internal exposure due to inhalation (mainly radon) and ingestion of naturally occurring radionuclides (K-40 and others) that adds to the mentioned external exposure, so that the total average annual dose is evaluated to be 2.4 mSv [271]. The Sv is the unit of equivalent dose, only applicable to stochastic effects, obtained by multiplying the unit of absorbed dose, Gy, by appropriate weighing factors to take into account the quality of radiation and the type of exposure; in the special case of uniform total-body exposure to low-LET radiation, 1 Sv = 1 Gy.

Despite the fact that the natural radiation background is presently extremely small, nevertheless it may be significant enough for living organisms to sense it and respond to it, keeping memory of this exposure. Changes in cell properties have been shown in bacterial, protozoan and mammalian cells cultured in low radiation environments such as those offered by underground laboratories [272,273,274,275,276,277,278]. Further experiments with a more complex organism, the fruit fly, indicated that reduction in radiation background significantly affected the fly lifespan and female fertility [279,280]. Overall, these experiments suggest that very low levels of chronic exposure, such as the natural background, may trigger defense mechanisms without genetic change, therefore by epigenetic mechanisms [278,279,280], an explanation that finds support in the already reviewed epigenetic origins of low-dose radiation responses, such as AR and NTE.

## 7. Concluding Remarks and Perspectives

### 7.1. Epigenetics Is Needed in Radiobiology Paradigms

Epigenetics is one of the fastest-growing areas of biological sciences, moving to the forefront of biomedical research, and also radiobiology could benefit from knowledge and control of epigenetic mechanisms. However, the involvement of epigenetic mechanisms in the biological response to ionizing radiation has not been studied as extensively as in other fields [11].

In contrast to conventional paradigms, the emerging picture of the cell response to ionizing radiation speaks in favor of a complex response to a variety of radiation-induced signals with perturbations at the cellular and supracellular levels, where epigenetic changes have become increasingly recognized as important aspects besides the genetic ones (Figure 4). Genetic and epigenetic mechanisms appear to have their common origin in radiation-induced ROS/RNS. This complex response is also the basis for the observed non-linear phenomena.

Knowledge about the basic radiobiological mechanisms is not only relevant to radiation biology, but it can also have a great impact on related applied science, notably in radiation protection. Indeed, it is essential for developing realistic models to guide extrapolations of epidemiological data on exposed human populations, so as to estimate risks at low doses and low dose-rates for both low- and high-LET radiation, and also to identify the factors determining individual radiosensitivity/susceptibility.

### 7.2. Implications in Radiation-Induced Cancer

The implication of epigenetic effects in radiation-induced cancer has not yet received much attention in developing mechanistic models of radiation action to be used for radiation protection purposes. It is interesting to note that, for example, ICRP [3] assumes that cancer development is best described as a multistep process originating from single cells that have sustained mutations through DNA damage and that, either directly or following the accumulation of additional mutations or epigenetic changes, such cells gain growth advantages and progress to a proliferative and ultimately malignant tumor. However, in partial contrast to this statement, radiation is then judged to act most commonly by inducing initiating mutations in proto-oncogenes or in tumor suppressor genes, an assumption that may overlook the contribution from epigenetic mechanisms. Since epigenetic changes have become increasingly recognized as important factors contributing to cancer development, models of radiation-induced carcinogenesis should be developed to estimate radiation risk with the incorporation of both genetic and epigenetic effects.

Moreover, epigenetic mechanisms may also have an impact on individual susceptibility to radiation-induced cancer. They may determine differences between individuals, or groups, and such differences, if significant, raise the ethical and policy question as to whether some individuals or groups are inadequately protected by the present system and regulations [253]. Therefore, research is required to clarify the role of epigenetic traits in order to settle this issue.

### 7.3. Implications in Radiation-Induced Hereditary Effects

Do transgenerational epigenetic effects impact human phenotypic variation and disease risk? To answer this question, it is necessary to resolve the discrepancies between human and animal/cellular data, so as to reach a consistent picture of this kind of effects. While clarifying this aspect will give insights about the mechanisms of this mode of inheritance, it is expected to have only a limited impact on radiation protection.

Indeed, it must be considered that in the current risk assessment, hereditary risks are only a minor contribution (about 4%) of the total detriment due to stochastic effects caused by ionizing radiation exposure, the major contribution being represented by cancer induction in the irradiated person, i.e., by somatic effects [1]. Therefore, a possible further contribution of transgenerational genome instability to the increase in mutation rates in the offspring of irradiated parents is likely not to affect the present radiation protection practice much, while it could be of great interest for improving our knowledge in radiation biology.

It is interesting to note that a relatively new problem would be represented by radiation therapy-related consequences. Although modern cancer radiotherapy has led to increased patient survival rates, the threat posed to the progeny of radiation-treated parents should be re-evaluated due to possible transgenerational carcinogenesis.

### 7.4. Implications in Radiation-Induced Non-Cancer Effects

In spite of the longstanding awareness of non-cancer somatic effects after medium/high doses of ionizing radiation, their induction after low/moderate doses is a relatively recent issue so that knowledge on their underlying biological mechanisms is poor [228]. It has been proposed that the mechanisms relevant to those effects in this range are likely different from those relevant at higher doses [240,281].

The possibility of a stochastic nature of these effects without dose thresholds raises a wide range of questions and may have important implications for radiation protection [253]. It is expected that epigenetic mechanisms are relevant to the future development of mechanistic models of radio-induced non-cancer diseases and, possibly, to the development of relevant markers in exposed individuals.

### 7.5. Low-Level Exposures: Detrimental or Beneficial?

In this review, many lines of evidence are described supporting the notion that cellular response to low dose of radiation are controlled, at least in part, by gene expression networks, and that epigenetic mechanisms are involved in adaptive response and in hormesis-like responses. While most of this information has been obtained in in vitro or ex vivo systems, it appears likely that adaptive epigenetic rearrangements can occur in human organisms, not only during early developmental stages but also throughout adulthood, improving their functional ability [267]. These effects have been attributed to the induction of the adaptive-response genes due to a long-lasting epigenetic memory in response to various kinds of mild stress [282].

While epidemiological data do not provide firm evidence for detrimental health effects below 100 mGy of low-LET radiation, human cellular responses to low doses of radiation that are typical of certain occupational activities or diagnostic radiography were often shown to harbor lower levels of chromosomal damage than that occurring spontaneously at the basal level ([283] and refs therein, [284] and refs therein). These considerations give some support to the assumption, based on studies with in vitro and animal models, that low-dose radiation has beneficial effects [285] and to the belief that LNT assumption (which implies proportionality between dose and health risk) is not valid at low doses or, at least, that it has not been proven to be true [284,286]. However, translating epigenetic-mediated cellular mechanisms, such as adaptive response, to the level of the human organism is not straightforward. Whether stimulation of cell defense mechanisms by low-level exposures is beneficial or deleterious in terms of health effects on human organism is not a trivial question. Although this stimulation is evocative of a positive reaction, it could not be the case if, for example, cells damaged by protracted exposures escape apoptosis, a situation that could enhance tumor promotion by increasing the probability of the survival of cells with accumulating damage or mutation [260]. Settling this controversy needs deeper insights of those radiobiological genetic and epigenetic mechanisms that dominate at low doses and at the same time are relevant to health effects on humans. In particular, there is a large consensus about the need for developing and using well-validated animal and human cellular/tissue models of radiation carcinogenesis [253].

Understanding the role of natural radiation background on living organisms is also essential to complete this scenario, given that it provides the biological background on which the response to man-made exposures overlap, and that this background shows large geographical variations [2]. It is expected that controlled long-term experiments with various model organisms, conducted in underground laboratories where conditions with extremely reduced background radiation are realized, can provide this basic information and, at the same time, can increase our knowledge about the role played by the natural radiation in life’s evolution.

Clearly, the decision whether the current LNT extrapolation of health risk estimates to low doses is still appropriate or whether a new paradigm has to be developed to provide more realistic protection against low radiation doses has significant social and economic implications.

Therefore, it is not surprising that quite often, the controversy is implicitly driven by considerations that are not strictly scientific but nevertheless can have a great impact on radiation protection practice. One of these considerations is that the LNT assumption makes the present system quite well manageable since a given dose can be a direct index of health risk and different doses received by an individual in different time periods can be summed up to evaluate the overall risk. On the other hand, no real alternative model based on recent scientific achievements has yet been proposed, likely because of the complex picture that has emerged for the biological response at low doses. However, these achievements suggest some general considerations useful in radiation protection practice such as, for example, that summing up many small doses to get an indication of the total health risk is unreasonable from the biological point of view, both at the individual and at the population level. This should be taken into account at least in performing the process called “optimization of the protection”, which is one of the three fundamental principles of radiation protection, so as to extend to this aspect the conclusion already expressed by the ICRP that the “collective effective dose is not intended as a tool for epidemiological studies, and it is inappropriate to use it in risk projections” [1], a reasonable conclusion when taking into account that the collective dose may be made of a sum of a large number of small individual doses.

## Figures and Tables

**Figure 1 ijms-21-05993-f001:**
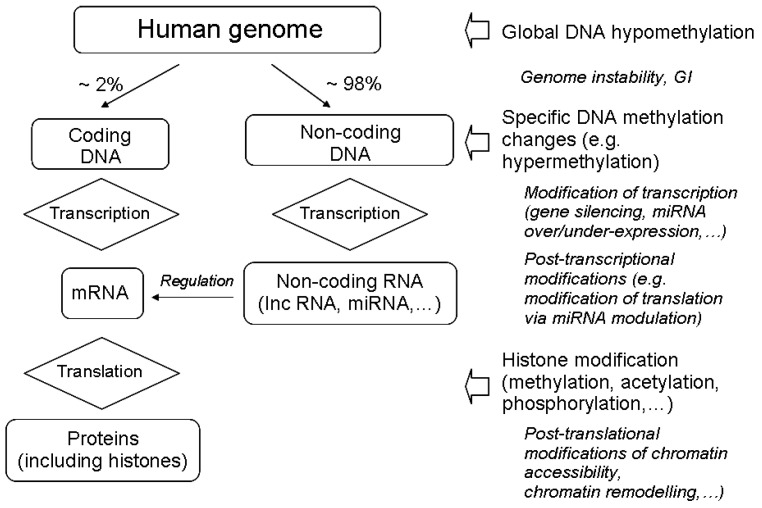
Epigenetic mechanisms involve different levels of gene expression (transcriptional, post-transcriptional, and post-translational). Only a small fraction of the human genome (2% or even less) accounts for protein-coding genes while the majority is associated with non-coding sequences, notably non-coding RNA genes. Of the non-coding DNA, only the regulatory part, giving rise to non-coding RNAs, is considered here. Epigenetic mechanisms can involve both protein-coding and non-coding RNA genes, with interplay between DNA methylation, histone modification and miRNA expression.

**Figure 2 ijms-21-05993-f002:**
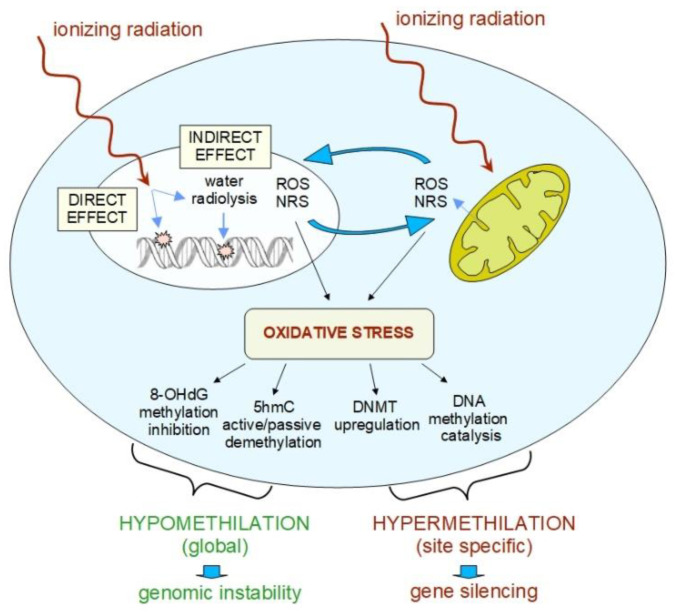
Simplified representation of the role of reactive oxygen species (ROS) and reactive nitrogen species (RNS) in the epigenetic response (DNA methylation) to ionizing radiation. Ionizing radiation can cause DNA lesions by direct deposition of energy in the DNA as well as by the indirect action of reactive chemical species formed near the DNA. Indirect DNA damage arises from free radicals formed through the radiolysis of water molecules. In aerobic conditions, these free radicals are converted to reactive oxygen species (ROS). Ionizing radiation can also generate reactive nitrogen species (RNS) through the upregulation of several enzymes. The yield and spatial distribution of ROS and RNS are strongly modulated by radiation quality because of the specific track structure of each quality [113]. ROS and RNS can attack DNA, resulting in several alterations that, if unrepaired or mis-repaired, may lead to genetic mutations in surviving cells. In addition, they can drive various epigenetic modifications through several mechanisms. It was demonstrated, especially in fibroblasts [114], that ROS can be directly generated by radiation exposure and indirectly through the radiation damage of mitochondria, leading to the activation of signaling pathways, which in turn sustains an increase in ROS levels. Oxidative stress results when excess of ROS/RNS are not compensated by the scavenging mechanisms of the cell. DNA hypomethylation can be achieved by oxidation of guanine to 8-OHdG that inhibits DNA methylation at nearby cytosine bases, and by hydroxylation of 5 mC to 5 hmC that causes active DNA demethylation processes. ROS can also induce site-specific DNA hypermethylation by up-regulation of expression of DNMTs or by acting as catalysts of DNA methylation [126]. Oxidative stress can influence the epigenetic landscape of the cell on other levels, such as by histone modifications and miRNA expression (not shown here).

**Figure 3 ijms-21-05993-f003:**
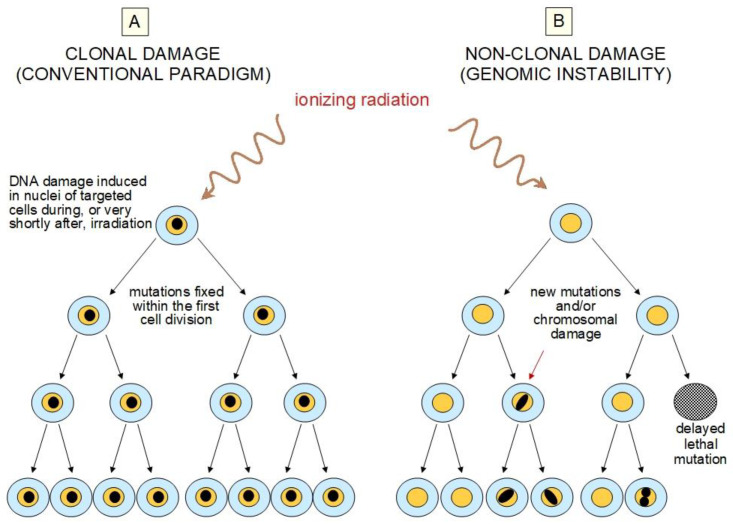
Schematic description of two possible ways for radiation-induced damage propagation in the progeny of irradiated cells: (**A**) according to the conventional paradigm of radiobiology, damage is induced during, or shortly after, irradiation, and clonally propagates to the progeny; (**B**) by genomic instability, a non-clonal effects that is observed as new mutations and/or new chromosomal damage in the unirradiated progeny of the irradiated cell.

**Figure 4 ijms-21-05993-f004:**
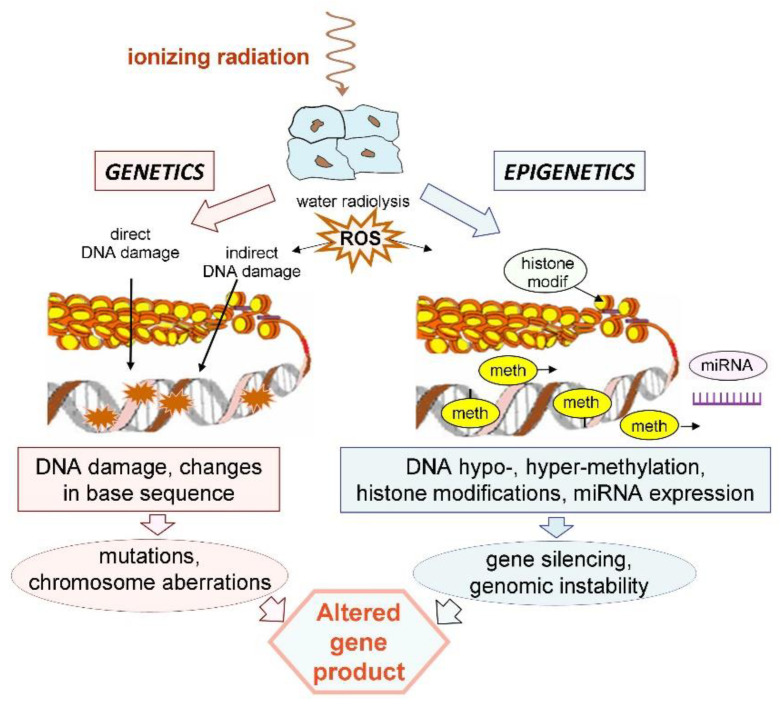
Schematic representation of the cell response to ionizing radiation, where alterations in gene products are due to both genetic and epigenetic mechanisms. These mechanisms share a common pathway originating from ROS production triggered by water radiolysis but, in addition, genetic changes can also be induced by DNA damaged via the direct action of radiation. Unbalanced ROS/RNS production results in oxidative stress with the involvement of mitochondria (not shown here).

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
