# Peer review of "Ionizing Radiation-Induced Epigenetic Modifications and Their Relevance to Radiation Protection"

_ijms, 2020, doi:10.3390/ijms21175993_

Round 1

Reviewer 1 Report

The authors in this rather extensive review target the epigenetic changes of obvious importance after radiation exposure. They discuss and cover many areas and the overall aim is very good and not really worked as a review by others recently, therefore it is relatively original and possibly deserves publication after revisions. The authors need to focus more on the real epigenetic effects and not really engage so much in Cancer or non cancer effects. Actually the epigenetic effect is considered a non cancer effect.

In addition the addition of Clustered DNA damage is not really covered to its correct and complete role and especially to its mechanistic importance triggering systemic effects and danger signals. Please look recent works and idea in Mavragani et al. Cancers (Basel).://pubmed.ncbi.nlm.nih.gov/31739493/. There it is documented that low and high LET radiations induce in many times a very different systemic effect, occurring probably as the result of the initial complex damage and intermediate responses including misrepair, differential epigenetic effects and others.

Another point that deserves attention is that the authors need to make clear what epigenetic changes we have for different radiation type, maybe a Table and what role is played by the organism. 

The connection between the DNA damage and epigenetic events must be better described. 

Overall this is a nice but very extensive review that must be more focused. 

Author Response

Please see the attachment "answer to Reviewer 1.pdf"

Reviewer 2 Report

This review article is informative and interesting. Please use higher resolution of the figures, and the author may provide one or two more figures for better comprehensive. 

Author Response

Reviewer 2

This review article is informative and interesting. Please use higher resolution of the figures, and the author may provide one or two more figures for better comprehensive. 

- Thanks for the warning about the resolution of figures. We increased it to the 300 dpi (minimum) requested by the publisher. We also added two more figures that help to explain in a graphical way some crucial points described in this review. Figure 2 (Simplified representation of the role of ROS and NRS in the epigenetic response) has been added to Sect.4 and Figure 3 (Schematic description of two possible ways for radiation-induced damage propagation in the progeny of irradiated cells) to Sect. 5. Another Reviewer also suggested increasing the number of figures.

Reviewer 3 Report

The manuscript brings important information regarding epigenetic changes induced by radiation. I recommend it for publication after minor queries are answered. They are the following:

First, some information are unnecessarily repeated throughout the text in different sections (besides those that are worth repeating). The authors should revise the text in order to avoid excessive repetitions.

Second, some sections have enough information to be shown also in figures, i.e. sections 4, 5, and 6). Overall, the manuscript is poor in figures and tables which could facilitate the understanding of the content.

Third, the epigenetic modifications are thoroughly described in the manuscript, however the "relevance to Radiation Protection" presented in the title is very briefly mentioned compared to the rest. Please reinforce it, as it is very important.

Lastly, some minor grammar mistakes and typos can be found throughout the text, therefore it needs English review.

Author Response

Please see the attachment "answer to Reviewer 3.pdf"

Reviewer 4 Report

Belli and Tabocchini have put together a quite comprehensive and useful review about ionizing radiation-induced epigenetic modifications and their relevance to radioprotection. The review seems lengthy but is well readable. As epigenetics are perceived to be important, this review is timely and addressing a very important aspect in biology and radiation biology.

The only shortcoming I have detected is the following:

It would be very helpful to mention time lines as how fast epigenetic changes occur – what has been found out – and discriminate these time lines and specific alterations in the different tissues and models examined.

Author Response

Please see the attachment "answer to Reviewer 4.pdf"
